# Effects of COVID-19 Financial and Social Hardships on Infants’ and Toddlers’ Development in the ECHO Program

**DOI:** 10.3390/ijerph20021013

**Published:** 2023-01-05

**Authors:** Sara S. Nozadi, Ximin Li, Xiangrong Kong, Brandon Rennie, Deborah Kanda, Debra MacKenzie, Li Luo, Jonathan Posner, Courtney K. Blackwell, Lisa A. Croen, Assiamira Ferrara, Thomas G. O’Connor, Emily Zimmerman, Akhgar Ghassabian, Leslie D. Leve, Amy J. Elliott, Rebecca J. Schmidt, Jenna L. N. Sprowles, Johnnye L. Lewis

**Affiliations:** 1Community Environmental Health Program, College of Pharmacy, University of New Mexico Health Sciences Center, Albuquerque, NM 87106, USA; 2Department of Biostatistics, Bloomberg School of Public Health, Johns Hopkins University, Baltimore, MD 21205, USA; 3Wilmer Eye Institute, School of Medicine, Johns Hopkins University, Baltimore, MD 21087, USA; 4Department of Pediatrics, Center for Development and Disability, University of New Mexico Health Sciences Center, Albuquerque, NM 87107, USA; 5Comprehensive Cancer Center, University of New Mexico, Albuquerque, NM 87106, USA; 6Department of Psychiatry & Behavioral Sciences, Duke University School of Medicine, Duke University, Durham, NC 27705, USA; 7Department of Medical Social Sciences, Feinberg School of Medicine, Northwestern University, Chicago, IL 60611, USA; 8Division of Research, Kaiser Permanente Northern California, Oakland, CA 94612, USA; 9School of Medicine and Dentistry, University of Rochester Medical Center, Rochester, NY 14642, USA; 10Department of Communication Sciences and Disorder, Northeastern University, Boston, MA 02115, USA; 11Department of Pediatrics, NYU Grossman School of Medicine, New York, NY 10016, USA; 12Department of Counseling Psychology, College of Education, University of Oregon, Eugene, OR 97403, USA; 13Department of Pediatrics, Sanford School of Medicine, University of South Dakota, Sioux Falls, SD 57105, USA; 14Avera Research Institute, Sioux Falls, SD 57108, USA; 15Department of Public Health Sciences, School of Medicine, University of California Davis, Davis, CA 95616, USA; 16Beckman Institute for Advanced Science and Technology, University of Illinois Urbana-Champaign, Urbana, IL 61801, USA; 17International Classification of Functioning, Disability, and Health, Durham, NC 27713, USA

**Keywords:** COVID-19, pandemic-related hardships, ages and stages questionnaire, ASQ, early development

## Abstract

Background: The financial hardships and social isolation experienced during the COVID-19 pandemic have been found to adversely affect children’s developmental outcomes. While many studies thus far have focused on school-aged children and the pandemic-related impacts on their academic skills and behavior problems, relatively less is known about pandemic hardships and associations with children’s development during their early years. Using a racially and economically diverse sample, we examined whether hardships experienced during the pandemic were associated with children’s development with a particular focus on communication and socioemotional development. Methods: Participants from eight cohorts of the Environmental influences on Child Health Outcomes program provided data on pandemic-related financial and social hardships as well as child developmental outcomes. Financial hardship was defined as at least one parent experiencing job loss or change, and social hardship was defined as families’ quarantining from household members or extended family and friends. The development of children under 4 was assessed longitudinally, before and during the pandemic (N = 684), using the Ages and Stages Questionnaire (ASQ). The Generalized Estimating Equations, which accounted for within-child correlation, were used for analysis. Results: Families from minority backgrounds and low socioeconomic status disproportionately experienced pandemic-related hardships. Male children had higher odds of experiencing negative changes in communication and personal social skills from pre- to during-pandemic visits (ORs ranged between 2.24 and 3.03 in analysis with binary ASQ outcomes and ranged from −0.34–0.36 in analyses with ASQ z-scores, *p*s = 0.000). Pandemic-related hardships in the social and financial areas did not explain within-individual changes in children’s developmental outcomes. Conclusion: Negative developmental changes from pre- to during-pandemic were found in boys, yet we did not find any associations between increased experience of pandemic-related hardships and children’s development. E how pandemic hardships affect development using a larger sample size and with longer follow-up is warranted.

## 1. Introduction

The emergence of COVID-19 in 2020 introduced disruption to children’s and families’ lives. Preventive guidelines and stay-at-home orders were put into place as early as March 2020 across most states [1]. Closures of schools, daycares and playgrounds or social distancing from friends and extended families in the United States had direct impacts on children’s lives by limiting their play, learning opportunities and social interactions with peers and adults [2]. Other pandemic-related disruptions, such as parents’ job loss and economic upheavals, indirectly affected children’s lives through disruptions to daily routines essential for optimal development during early years [3,4,5]. Two developmental domains that likely have been most affected by pandemic-related disruptions are children’s cognitive and socioemotional development [4]. Positive impacts also have been noted as the result of preventive guidelines (e.g., working from home) such as an increase in time that family members spend together and potential for bonding among family members as well as an increase in time that children spend in nature, which has been associated with positive impacts on development [6,7]. The increase in bonding time between parent and their children as the result of working-from-home practices can be particularly important for young children as the first three years are important in the development of attachment [8].

While numerous studies have focused on understanding the impact of the pandemic on children’s developmental outcomes, few studies have included children under the age of five. Studies that have examined developmental outcomes among children under age 5 have either used cross-sectional data to evaluate differences in the population-trends between before and during the pandemic or have examined the effects of in utero exposure to COVID-19 on infants’ developmental outcomes [5,9,10]. A recent study compared data on cognitive assessments collected prior to the pandemic from children between 3 months and 3 years of age to data from same-aged children of the same cohort collected during the first year of the pandemic [9]. Results showed that children’s scores on cognitive measures collected during the pandemic were significantly lower than scores in the pre-pandemic group, mainly driven by a small group of 39 children who were born right before or during the pandemic. In another cohort study, researchers compared the developmental outcomes of 6-month-old infants, assessed via ASQ [11], and observed that infants born during the pandemic had lower developmental scores on motor and personal-social scores than 6-month-old children assessed prior to the pandemic [10]. Both of these studies utilized cross-sectional data to examine developmental outcomes rather than examining children’s development across time. To address this gap in the literature, we examined the effects of pandemic-related hardships on developmental changes of children between 2 and 48 months of age in a diverse sample of U.S. children, while considering baseline developmental status and sociodemographic characteristics known to be strong predictors of children’s development. Given the high dependency that children have on caregivers during the first few years of life, they may be more vulnerable to parental stress and negative family dynamics, which may impact the quality and frequency of positive parent–child interactions [4]. Thus, we were particularly interested in evaluating the associations between pandemic-related hardships and young children’s communication and socioemotional development.

Study participants were part of the ECHO program [12]. We focused on the social and financial hardships reported by parents including job changes or loss and social distancing from family members and friends because these situations may limit children’s learning opportunities and/or parents–child interactions through an increase in parental stress [9,13]. We hypothesized that financial hardships and disturbances experienced in socialization practices during the pandemic would be associated with children’s loss in developmental functioning, particularly in the areas of communication and socioemotional development.

## 2. Materials and Methods

### 2.1. Study Population

Participants were families from eight cohorts within the ECHO program who had (1) data on child ASQ within the 18-month period before and after March 2020, (2) parent-reported data on hardships experienced in either financial or social areas, and (3) had a child ≤48 months at the most recent study visit prior to March 2020. For participants who had multiple pre-pandemic ASQ scores, the scores from the most recent visit were used. All available ASQ data collected during the pandemic were considered, when child had multiple assessments. Figure 1 summarizes the inclusion criteria considered in determining the final sample.

A total of 684 participants from eight ECHO cohorts contributed data to this analysis. Most participants resided in the Northeast (60.53%) with 26.61% living in the West, 10.23% living in the Midwest, and 2.63% living in the South (based on region designation by the U.S. Census Bureau). Ninety-one participants (~13.3% of the sample) did not have information on the residential address so the study site was used to determine region. We considered only the pre-pandemic ASQ data that were collected within the 18-month window prior to March 2020 (i.e., August 2018–11 March 2020) to match the time frame for the ASQ data included in the analysis from the during-pandemic period (March 2020–August 2021).

### 2.2. Measures

Primary Exposures: COVID-Related Financial and Social Hardships. The pandemic-related hardships were assessed using four items from the COVID-19 survey, developed by the ECHO Consortium [14]. The ECHO COVID-19 survey was developed in April 2020 by a group of ECHO investigators with various expertise (e.g., clinical and developmental psychologists, epidemiologists) using existing surveys and source materials from leading organizations. More detailed information about the development process can be found in Blackwell et al. (2022) article [15]. Two items were used to assess financial hardships: “In what ways has the COVID-19 outbreak affected your (your spouse’s) work?” We defined negative financial influences as the experience of at least one parent losing their job, temporarily or permanently, or experiencing reduced work hours (e.g., “I (my spouse/partner) lost job temporarily, or was not told for how long.”).

Two additional items assessed negative influences of the pandemic on social life defined as mothers’ report of less contact with household members (i.e., “quarantined separately from one or more family or household members”), or less contact with family and friends outside the home. Scores for each hardship category ranged from 0–2 with higher numbers reflecting more hardships. The items used to capture financial and social hardships were similar to those from other well-established COVID-19 surveys such as the COVID-19 Exposure and Family Impact Survey (CEFIS) [16].

Developmental Outcomes. Children’s developmental scores were collected using the Ages and Stages Questionnaires™, third edition (ASQ-3) [17,18], a parent-reported and widely used developmental screener consisting of 21 age-specific questionnaires from 2 months to 60 months of age. Child’s development is assessed across five domains: communication, gross motor, fine motor, problem-solving, and personal-social. Each domain consists of six questions asking the caregivers to rate whether the child has acquired a developmental skill (10 = “Yes”; 5 = “Sometimes”; 0 = “Not Yet”). The obtained score was the sum of all six questions with higher scores indicating more advanced developmental level. Final scores were compared to cut-offs established in the ASQ normative sample to assign a developmental status indicating that the child was (1) developing normally (i.e., “on track”), (2) in need of further monitoring (i.e., “needs monitoring”) or (3) in need for further comprehensive evaluation (i.e., “at-risk for delay”) [17,18,19]. For preterm children with gestational age <36 weeks, adjusted age was used to determine the appropriate ASQ-3 questionnaire. The majority of ASQs were administered in English (65.4%), with 11.4% administered in Spanish, and 23.2% were missing information about the ASQ administration language. The language format used for administration changed between pre- and during-the pandemic for 0.7% of children.

In addition to categories for developmental status in each domain, the domain-specific ASQ scores were converted to z-scores using the mean and standard deviation (SD) from the normative sample. The z-scores are the normalized ASQ-score with respect to the corresponding age-specific ASQ from the normative sample, and thus can be interpreted as the deviation from normal of that age. The z-scores were calculated as follows:

(Observed ASQ score- mean of the corresponding age-specific ASQ obtained from the normative sample)/SD of the normative sample.

Outcome Measures. To evaluate change in developmental level over time, two outcome measures based on the ASQ were defined and analyzed separately for each domain. The first variable was a risk index indicating whether the child’s developmental status indicated loss in developmental functioning (i.e., changed from “on track” to “needs monitoring” or “at-risk for delay” status, changed from “needs monitoring” to “at-risk for delay” status). The second variable that was considered in the analysis was the change in z-scores from pre- to during-pandemic visits. There were 212 out of 684 children with multiple assessments during the pandemic (Mean = 1.40, Median = 1). The number of children with gain, loss, and no change in their developmental categories (status) included (1) 105, 147, and 696 children for the communication domain, (2) 131, 106 and 711 children for the gross motor domain, (3) 100, 79, and 478 children for the fine motor, 99, 64, and 496 children for the problem-solving domain, and 2, 82, and 496 children for the personal-social domain, respectively.

Covariates. Covariates included were maternal race, maternal ethnicity, maternal education, household income, marital status, preterm birth, and children’s sex, race, ethnicity, gestational age, age at the pandemic visit (2–12 months = infancy versus 12–48 months = toddlerhood) and baseline neurodevelopment (i.e., pre-pandemic ASQ z-scores). Maternal and child race/ethnicity were modeled in a separate analysis. For maternal education, household income and marital status, information collected at the time of the pre-COVID ASQ visit were used, when available. When not available, data from the prenatal visit were used, which represented 77.5% of maternal education, 55.2% of household income, and 78.1% of marital status data points. Two other covariates not reported due to high missingness included ASQ administration mode (self-administered vs. staff-administered) and ASQ administration setting (in clinic/site, by phone, or at another location).

### 2.3. Statistical Analyses

Demographic and pre-pandemic descriptive statistics were summarized. *t*-tests and chi-square tests were used for continuous and categorical covariates, respectively, to compare families who reported pandemic-related financial or social hardships versus those who did not report any hardships. To deal with missing data for the covariates, multiple imputations with Chained Equation (10 imputations with 10 iterations) was used to impute missing data for variables of maternal race, ethnicity, maternal education (5 categories), maternal employment (2 categories), household income (5 categories), marital status, preterm, gestational age, children sex, race, ethnicity, and birth weight.

As some children had multiple ASQ scores during the pandemic, we used the Generalized Estimating Equation (GEE) with exchangeable working correlations accounting for within-child correlation. Univariate analysis was performed for each ASQ domain for each of the exposure variable, followed by multivariate GEE adjusting for baseline ASQ z-scores and covariates that had *p* value smaller than 0.2 in univariate analysis. Specifically, for the outcome of the risk index indicating loss in developmental functioning, logistic regression models with GEE were used; and for the outcome of the change in ASQ z-scores, linear models with GEE were used. As there were multiple primary exposure variables and the outcome of ASQ change was assessed in two ways, statistical significance was considered as 0.008 to account for multiple comparisons (3 primary exposures, 2 ways for each domain) using the Bonferroni method.

## 3. Results

### 3.1. Preliminary Results

Figure 1 shows the flowchart of the current study sample. A total of 684 children met the criteria for inclusion in this study. Pre-pandemic (baseline) characteristics of the study sample including families’ sociodemographic and child characteristics are summarized in Table 1. Among 684 children, 48% of children were <12 months at the time of pre-pandemic assessment and 47.4% were female. Most mothers in the sample were non-Hispanic (68.7%) and 50.9% of the mothers were White. About 62% of mothers in the sample had either a bachelor’s or graduate degree, and about 71% of mothers were employed. Further, 47.7% of families reported an annual income of $100,000 or more and 88.7% of mothers were married. More than half of parents (53%) reported having experienced financial hardships, whereas 88.8% of parents reported less contact with family and friends outside the home.

### 3.2. Pandemic-Related Hardships by Pre-Pandemic Characteristics

Table 1 shows pandemic-related financial or social hardships by participants’ pre-pandemic sociodemographic and child characteristics. Maternal race, ethnicity, and education as well as the child’s ethnicity and family annual income were significantly associated with experiencing pandemic-related financial hardships. For example, financial hardships were more likely to be reported by Hispanic/Latino (77.9%) than non-Hispanic/Latino (43.2%) mothers, or those who had some college education (with or without an associate degree; 63.7%) compared to mothers with bachelor’s degrees (38.5%).

Maternal race and education were significantly associated with parents reporting less contact with household members. For example, whereas 9.2% of white mothers reported quarantining from household members, percentages were generally higher among non-white mothers (e.g., 40% of Indigenous mothers reported quarantining from household members). Not being married or living with a partner and having a mid-level household income were both associated with less in-person contact with family members/friends outside the home. As shown in Table 1, the distributions of children’s pre-pandemic ASQ developmental status were not significantly associated with whether their parents reported pandemic-related financial or social hardships.

Loss in developmental functioning from prior to during the pandemic was observed for 8–15% of children depending on the domain:15.3% (communication), 11.4% (gross motor), 11.8% (fine motor), 8.0% (problem-solving), 11.0% (personal-social).

The adjusted odds ratios estimated for loss in developmental functioning for communication and personal-social domains are presented in Table 2; the unadjusted odds ratios estimated from univariate regression models for all domains are presented in supplemental Appendix A. For the communication and personal-social domains, none of the three primary exposure variables nor covariates were significantly associated with worsening in the ASQ category, except for child sex. Male children were more than twice as likely to evidence loss in developmental functioning in the communication (OR = 2.4 [95% CI 1.61–4.00], *p* < 0.001) and personal-social (OR = 2.4 [95% CI: 1.46–4.00], *p* < 0.001) domains compared to female children; the adjusted ORs from multivariate models were similar.

Results from the multivariate models for the remaining ASQ domains are presented in Appendix A. For the gross motor and problem-solving domains, no significant results were observed. For fine motor, maternal education was significantly associated with ASQ worsening in univariate and multivariable models. For example, compared to children of mothers with college degrees, children of mothers with less than high school and those with high school or equivalent education had higher odds of reporting less contact with household members, ORs = 8.66 and 5.40 (95% CIs: 2.61–29.00 and 1.71–17.00, *p*s < 0.001), respectively. These ORs are of similar levels in other multivariable models. We conducted post hoc sensitivity analyses, controlling for ASQ outcomes as binary scores in all aforementioned analysis with similar findings.

### 3.3. Associations between Pandemic-Related Hardships and Change in ASQ Z-Scores 

From the pre- to during-pandemic visits, the means (SDs) of change in ASQ z-scores (during the pandemic score subtracted from pre-pandemic score) were −0.10 (1.17), 0.05 (1.14), 0.08 (1.26), 0.22 (1.20), −0.02 (1.14) for the domains of communication, gross motor, fine motor, problem solving and personal–social, respectively. Table 3 shows the results from multivariate regression models to identify exposures and covariates associated with a change in ASQ z-scores for the communication and personal-social domains. The regression results from univariate models are presented in Appendix A, and results from a multivariate analysis involving the other three domains are listed in Appendix A.

The three primary exposure variables on parent reports of pandemic-related financial or social hardships were not associated with a change in z-score in any domain regardless of the baseline ASQ scores. However, higher pre-pandemic ASQ z-scores were associated with a more negative change in the z-score during the pandemic (*p*-values all <0.001). For the communication domain, male children compared to female children and toddlers compared to infants had larger negative changes in z-scores (*p*-values all <0.001) in all of the unadjusted and adjusted models. Similarly, for the personal-social domain, male children had a more negative change in z-scores (*p*-values all <0.001) in all of the unadjusted and adjusted models.

## 4. Discussion

This study investigated changes in infants’ and toddlers’ development using a commonly used clinical screening tool (ASQ), from before to during the pandemic within a large and diverse sample using repeated measures and a longitudinal design. Consistent with previous studies, we found that families from racial and ethnic minority backgrounds and lower SES were more likely to report social and financial hardships during the COVID-19 pandemic [20]. However, these hardships did not explain within-individual changes in children’s development using either categorical or continuous outcomes.

Approximately 8–15% of children experienced a loss in developmental functioning from pre- to during- the pandemic, with male children at more than twice the risk of showing such loss as female children. This association was also robustly observed using the ASQ z-score changes as the outcome variables. These results were aligned with those from previous studies showing that male children are at increased risk for several neurodevelopmental disorders, including autism and language disorders, which suggest a general developmental vulnerability compared to girls [21,22,23]. These results also highlight the importance of improving service deliveries and early developmental screening during the well-child care visits at the pediatric care offices particularly at the time of adversity to ensure that young children who are at-risk including male children are connected to intervention and treatment services.

Other significant predictors of negative change in our analysis, although not as robustly observed across different outcomes, involved maternal education and baseline ASQ scores. The positive association between maternal education and fine motor skills has been reported in previous studies [24]. Children with higher ASQ baseline scores were also more at risk of having a negative change, which was expected and suggestive of regression toward the means [25].

Many studies in the U.S. have reported learning losses in the areas of reading and math from pre- to post-pandemic among school-aged children [26,27,28]. The results of a national study also showed that the achievement gaps in the areas of reading and math between students in low- and high-poverty schools significantly increased from pre- to post-pandemic years (difference of 0.10–0.20 SDs) [26]. In the current study, we did not find any associations between financial and social disruptions caused by the pandemic and infants’ and toddlers’ development, measured by an early screening tool. Given that early childhood developmental needs are different from those of older and school-aged children, future research may consider focusing on other aspects of pandemic-related disturbances such as the pandemic effects on family dynamics, parent–child interactions and parental stress. Additionally, more research is warranted to examine the effects of pandemic-related hardships and COVID-19 infection on children’s later developmental outcomes using a longer follow-up. The difference in the impact of disruptions related to the pandemic for younger and older children may also reflect the possibility that classroom-based learning platforms and achievement indices in the U.S. are more sensitive to disruptions than learning that takes place in the home environment. If further research studies find these differences, there may be a need to revise our traditional learning modes in the school context, which may lead to greater flexibility and adaptability to future hardships and disruption both on a national scale, as in the case of COVID, and on a state, community, and even individual student level.

## Figures and Tables

**Figure 1 ijerph-20-01013-f001:**
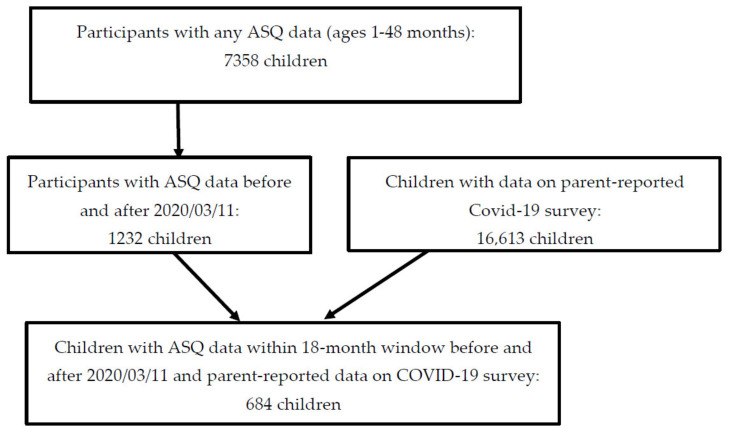
Inclusion criteria considered for determining sample sizes.

**Table 1 ijerph-20-01013-t001:** Characteristics of pooled sample at Baseline.

		Influence of Job Loss/Change			Less Contact with Household Members			Less Contact with Family/Friends Outside the Home		
	Total	No	Yes	*p*	No	Yes	*p*	No	Yes	*p*
N	684	247	279		580	96		76	600	
Maternal Characteristics										
**Maternal race (%)**				<0.001			<0.001			0.016
White	348 (50.9)	150 (54.7)	124 (45.3)		314 (90.8)	32 (9.2)		30 (8.7)	316 (91.3)	
Black	27 (3.9)	7 (28)	18 (72)		<28	<5		5 (18.5)	22 (81.5)	
Asian	84 (12.3)	35 (57.4)	26 (42.6)		67 (81.7)	15 (18.3)		8 (9.8)	74 (90.2)	
Native Hawaiian or Pacific Islander	7 (1)	<5	<7		7 (100)	0 (0.0)		0 (0.0)	7 (100)	
American Indian/Alaska Native	57 (8.3)	26 (53.1)	23 (46.9)		33 (60)	22 (40)		12 (21.8)	43 (78.2)	
Multiple race	33 (4.8)	15 (60)	10 (40)		27 (81.8)	6 (18.2)		7 (21.2)	26 (78.8)	
Other race	111(16.2)	9 (12.5)	63 (87.5)		97 (89)	12 (11)		9 (8.3)	100 (91.7)	
Missing	17 (2.5)	<5	<13							
**Maternal ethnicity (%)**				<0.001			0.392			1
Hispanic or Latino	214 (31.3)	33(22.1)	116 (77.9)		186 (87.7)	26 (12.3)		24 (11.3)	188 (88.7)	
Not Hispanic or Latino	470 (68.7)	214 (56.8)	163 (43.2)		394 (84.9)	70 (15.1)		52 (11.2)	412 (88.8)	
**Maternal education (%)**				<0.001			0.001			0.009
Less than high school	40 (5.8)	<5	<20		<36	<5		5	34	
High school degree	82 (12)	9 (15)	51 (85)		71 (86.6)	11 (13.4)		14 (17.1)	68 (82.9)	
Some college	126 (18.4)	37 (36.3)	65 (63.7)		94 (75.2)	31 (24.8)		23 (18.4)	102 (81.6)	
Bachelor’s (BA,BS)	187 (27.3)	88 (61.5)	55 (38.5)		158 (84.9)	28 (15.1)		17 (9.1)	169 (90.9)	
Graduate degree	237 (34.6)	107 (56)	84 (44)		214 (91.5)	20 (8.5)		17 (7.3)	217 (92.7)	
Missing	12 (1.8)	<5	<7		<10	<5				
**Maternal employment**				0.708			0.114			0.011
Employed	484 (70.8)	186 (50)	186 (50)		426 (89.1)	52 (10.9)		45 (9.4)	433 (90.6)	
Not employed	26 (3.8)	11 (44)	14 (56)		20 (76.9)	6 (23.1)		7 (26.9)	19 (73.1)	
Missing	174 (25.4)									
**Household income (%)**				<0.001			0.593			<0.001
<$30,000	92 (13.5)	8 (11.6)	61 (88.4)		78 (85.7)	13 (14.3)		15 (16.5)	76 (83.5)	
$30,000–49,999	45 (6.6)	9 (29)	22 (71)		<44	<5		6 (13.3)	39 (86.7)	
$50,000–74,999	52 (7.6)	22 (45.8)	26 (54.2)		44 (84.6)	8 (15.4)		12 (23.1)	40 (76.9)	
$75,000–99,999	48 (7)	18 (47.4)	20 (52.6)		38 (82.6)	8 (17.4)		7 (15.2)	39 (84.8)	
$100,000 or more	326 (47.7)	156 (61.2)	99 (38.8)		288 (88.9)	36 (11.1)		18 (5.6)	306 (94.4)	
Missing	121 (17.7)									
**Marital status**				0.222			0.25			0.004
Married/living with partner	607 (88.7)	226 (47.9)	246 (52.1)		519 (86.4)	82 (13.6)		61 (10.1)	540 (89.9)	
Not married	66 (9.6)	18 (37.5)	30 (62.5)		53 (80.3)	13 (19.7)		15 (22.7)	51 (77.3)	
Missing	11 (1.6)									
	**Child Characteristics**									
**Preterm**				1			1			1
37+ wks	620 (90.6)	223 (46.4)	258 (53.6)		528 (86)	86 (14)		66 (10.7)	548 (89.3)	
<37 wks	53 (7.7)	17 (47.2)	19 (52.8)		45 (86.5)	7 (13.5)		6 (11.5)	46 (88.5)	
Missing	11 (1.6)									
**Children sex (%)**				0.451			0.506			0.72
Female	324 47.4)	110 (44.9)	135 (55.1)		278 (86.9)	42 (13.1)		38 (11.9)	282 (88.1)	
Male	359 (52.5)	136 (48.6)	144 (51.4)		301 (84.8)	54 (15.2)		38 (10.7)	317 (89.3)	
**Children ethnicity (%)**				<0.001			0.249			0.798
Hispanic or Latino	229 (33.5)	42 (25.8)	121 (74.2)		199 (87.7)	28 (12.3)		25 (11)	202 (89)	
Not Hispanic or Latino	430 (62.9)	195 (56.9)	148 (43.1)		356 (84)	68 (16)		51 (12)	373 (88)	
Missing	25 (3.7)									
**Children race (%)**				0.5			0.4			0.8
White	349 (51)	135 (50)	135 (50)		312 (89.9)	35 (10.1)		30 (8.6)	317 (91.4)	
Black	23 (3.4)	<5	<16		<23	<5		<5	<23	
Asian	63 (9.2)	28 (59.6)	19 (40.4)		49 (80.3)	12 (19.7)		8 (13.1)	53 (86.9)	
Native Hawaiian/Pacific Islander	5 (0.7)	<5	<5		8 (13.1)	0 (0.0)		0 (0.0)	5	
American Indian	59 (8.6)	27 (54)	23 (46)		35 (61.4)	22 (38.6)		13 (22.8)	44 (77.2)	
Multiple race	75 (11)	33 (56.9)	25 (43.1)		64 (85.3)	11 (14.7)		9 (12)	66 (88)	
Other race	0 (0)	0 (0.0)	0 (0.0)		0 (0.0)	0 (0.0)		0 (0.0)	0 (0.0)	
Missing	110 (16.1)									
**Gestational age/birth weight (ga/bw)**										
Birth_ ga (mean (SD))		38.77 (1.59)	38.63 (2.78)	0.489	38.61 (2.27)	38.82 (1.47)	0.398	38.58 (1.96)	38.65 (2.20)	0.815
Birth_ bw (mean (SD))		3371.52 (509.87)	3332.98 (520.79)	0.405	3306.40 (518.48)	3379.07 (509.01)	0.221	3320.47 (619.27)	3315.80 (504.49)	0.944
**Children Pre-pandemic Life Stage (%)**	684 (100)			1			0.086			0.6
Early childhood	356 (52)	129 (46.9)	146 (53.1)		292 (83.4)	58 (16.6)		42 (12)	308 (88)	
Infancy	328 (48)	118 (47)	133 (53.0)		288 (88.3)	38 (11.7)		34 (10.4)	292 (89.6)	
	**Ages and Stages Questionnaire (ASQ)**									
**Communication (%)**				0.075			0.243			0.679
On-track		197 (45.2)	239 (54.8)		476 (85.2)	83 (14.8)		63 (11.3)	496 (88.7)	
Needs monitoring		33 (50.8)	32 (49.2)		<80	<11		<9	78 (100)	
At-risk for delay		14 (70)	6 (30)		<23	<5		<5	22 (100)	
**Gross motor (%)**				0.819			0.896			0.988
On-track		202 (47)	228 (53)		478 (86.1)	77 (13.9)		62 (11.2)	493 (88.8)	
Needs monitoring		27 (45)	33 (55)		67 (87)	10 (13)		<10	68	
At-risk for delay		12 (41.4)	17 (58.6)		31 (83.8)	6 (16.2)		<5	33	
**Fine motor (%)**				0.186			0.347			0.059
On-track		167 (47.7)	183 (52.3)		368 (84.6)	67 (15.4)		46 (10.6)	389 (89.4)	
Needs monitoring		20 (37.7)	33 (62.3)		61 (88.4)	8 (11.6)		11 (15.9)	58 (84.1)	
At-risk for delay		19 (57.6)	14 (42.4)		32 (78)	9 (22)		9 (22)	32 (78)	
**Problem solving (%)**				0.744			0.846			0.965
On-track		162 (46.3)	188 (53.7)		372 (84.4)	69 (15.6)		54 (12.2)	387 (87.8)	
Needs monitoring		30 (51.7)	28 (48.3)		55 (85.9)	9 (14.1)		8 (12.5)	56 (87.5)	
At-risk for delay		15 (46.9)	17 (53.1)		36 (81.8)	8 (18.2)		6 (13.6)	38 (86.4)	
**Personal social (%)**				0.063			0.213			0.126
On-track		164 (46.7)	187 (53.3)		376 (85.1)	66 (14.9)		48 (10.9)	394 (89.1)	
Needs monitoring		34 (56.7)	26 (43.3)		55 (77.5)	16 (22.5)		13 (18.3)	58 (81.7)	
At-risk for delay		8 (29.6)	19 (70.4)		27 (79.4)	7 (20.6)		6 (17.6)	28 (82.4)	

Note. Values indicated in bold presents those with significance values greater than 0.008. Associations Between Pandemic-Related Hardships and Change in ASQ Developmental Status (Categories).

**Table 2 ijerph-20-01013-t002:** Results of Multivariate Models Testing for the Effects of COVID-Related Hardships on ASQ Communication and Personal-Social Categories.

	Communication						Personal-Social					
Exposure Variables	Parents’ Job Loss/Change		Less Contact with Household Members		Less Contact with Family/Friends Outside Home		Parents’ Job Loss/Change		Less Contact with Household Members		Less Contact with Family/Friends Outside Home	
	OR (95% CI)	*p* Value	OR (95% CI)	*p* Value	OR (95% CI)	*p* Value	OR (95% CI)	*p* Value	OR (95% CI)	*p* Value	OR (95% CI)	*p* Value
Parents’ job loss/change	0.66 (0.37,1)	0.14					1.15 (0.63,2)	0.66				
less contact with household members			0.92 (0.49,2)	0.80					0.67 (0.34,1)	0.26		
Less contact with family/friends outside home					2.03 (0.91,5)	0.08					1.23 (0.58,3)	0.59
Pre-pandemic ASQ	0.72 (0.54,1)	0.02	0.98 (0.81,1)	0.82	0.97 (0.81,1)	0.76	0.87 (0.64,1)	0.39	1.12 (0.92,1)	0.24	1.13 (0.93,1)	0.21
**Maternal/family characteristics**												
American Indian or Alaskan Native	0.56 (0.17,2)	0.33	0.55 (0.21,1)	0.22	0.56 (0.22,1)	0.23	1.49 (0.52,4)	0.46	1.22 (0.48,3)	0.68	1.08 (0.43,3)	0.87
Asian	0.6 (0.23,2)	0.29	0.84 (0.43,2)	0.62	0.83 (0.41,2)	0.58	1.41 (0.6,3)	0.44	1.55 (0.76,3)	0.23	1.48 (0.72,3)	0.29
Black	0.88 (0.19,4)	0.87	0.97 (0.29,3)	0.96	0.95 (0.27,3)	0.94	1.43 (0.37,5)	0.60	1.21 (0.35,4)	0.76	1.21 (0.35,4)	0.76
Multiple Race	0.97 (0.31,3)	0.96	1.28 (0.51,3)	0.59	1.34 (0.53,3)	0.53	1.32 (0.37,5)	0.67	1.48 (0.51,4)	0.47	1.47 (0.51,4)	0.48
Native Hawaiian or other Pacific Islander	1.3 (0.28,6)	0.74	1.64 (0.38,7)	0.51	1.53 (0.36,7)	0.56	6.1 (0.62,60)	0.12	1.84 (0.22,16)	0.58	1.86 (0.22,16)	0.57
Other Race	1.05 (0.39,3)	0.92	1.02 (0.47,2)	0.97	0.98 (0.46,2)	0.97	0.94 (0.4,2)	0.89	0.94 (0.48,2)	0.84	0.93 (0.48,2)	0.84
Maternal ethnicity	0.73 (0.31,2)	0.46	0.85 (0.45,2)	0.60	0.87 (0.46,2)	0.67						
<$30,000	1.85 (0.52,7)	0.34	1.97 (0.82,5)	0.13	2.04 (0.85,5)	0.11						
$30,000–49,999	0.98 (0.31,3)	0.98	1.1 (0.43,3)	0.84	1.13 (0.44,3)	0.79						
$50,000–74,999	2.16 (0.9,5)	0.09	1.8 (0.82,4)	0.14	1.97 (0.89,4)	0.09						
$75,000–99,999	1.08 (0.38,3)	0.88	0.95 (0.39,2)	0.90	0.98 (0.41,2)	0.97						
Less than high school	2.61 (0.73,9)	0.14	1.9 (0.72,5)	0.20	2.05 (0.77,5)	0.15						
High school degree, GED or equivalent	2.89 (0.92,9)	0.07	1.81 (0.74,4)	0.19	1.92 (0.77,5)	0.16						
Some college, no degree, or AA	1.18 (0.46,3)	0.73	1.07 (0.5,2)	0.87	1.12 (0.51,2)	0.78						
Master’s degree, professional or doctorate degree	0.79 (0.41,2)	0.48	0.72 (0.41,1)	0.26	0.73 (0.41,1)	0.28						
**Child characteristics**												
Children sex	3.03 (1.75,5)	0.00	2.55 (1.65,4)	0.00	2.54 (1.64,4)	0.00	2.23 (1.22,4)	0.01	2.25 (1.35,4)	0.00	2.24 (1.34,4)	0.00
Gestational age							0.97 (0.84,1)	0.65	1.05 (0.87,1)	0.59	1.05 (0.87,1)	0.63
Preterm status							6.18 (0.68,56)	0.10	1.67 (0.45,6)	0.44	1.61 (0.45,6)	0.46
Children pre-pandemic life stage							−0.69 (−0.38,−1)	.22	−0.72 (−0.43,−1)	0.22	−0.71 (−0.43,−1)	.20

Notes. OR = Odds Ratio and CI = Confidence Interval; OR and CI values indicated in bold present those with significance values greater than 0.008. References groups for categorical variables were White (maternal race), college degree (maternal education), Hispanic (maternal ethnicity), female (child’s sex), and infancy (pre-pandemic stage). All models were adjusted for pre-pandemic ASQ z-scores and significant covariates (*p*-values < 0.2).

**Table 3 ijerph-20-01013-t003:** Results of Multivariate Models Testing for the Effects of COVID-Related Hardships on ASQ Communication and Personal-Social Z-Scores.

	Communication						Personal-Social					
Variables	Parents’ Job Loss/Change		Less Contact with Household Members		Less Contact with Family/Friends Outside Home		Parents’ Job Loss/Change		Less Contact with Household Members		Less Contact with Household Members	
	Estimate (95% CI)	*p* Value	Estimate (95% CI)	*p* Value	Estimate (95% CI)	*p* Value	Estimate (95% CI)	*p* Value	Estimate (95% CI)	*p* Value	Estimate (95% CI)	*p* Value
Parents’ job loss/change	0.12 (−0.04,0.27)	0.15					−0.02 (–0.2,0.17)	0.85				
Less contact with household members			0.1 (−0.11,0.31)	0.36					0.15 (−0.06,0.36)	0.17		
Less contact with family/friends outside home					−0.14 (−0.37,0.09)	0.22					−0.18 (−0.43,0.08)	0.17
Baseline ASQ	−0.65 (−0.75,−0.54)	0.00	−0.63 (−0.72,−0.54)	0.00	−0.63 (−0.72,−0.54)	0.00	−0.56 (−0.67,−0.46)	0.00	−0.56 (−0.67,−0.46)	0.00	−0.57 (−0.67,−0.46)	0.00
**Maternal/family characteristics**												
American Indian or Alaska Native	0.25 (−0.17,0.67)	0.24	0.22 (−0.15,0.59)	0.25	0.24 (−0.12,0.59)	0.19	−0.07 (−0.48,0.35)	0.75	0.01 (−0.39,0.4)	0.96	0.05 (−0.33,0.43)	0.79
Asian	0.06 (−0.16,0.28)	0.57	−0.02 (−0.22,0.19)	0.87	−0.01 (−0.22,0.2)	0.93	0.02 (−0.27,0.31)	0.89	−0.06 (−0.33,0.21)	0.68	−0.04 (−0.31,0.23)	0.77
Black	−0.01 (−0.46,0.45)	0.98	−0.01 (−0.43,0.41)	0.98	−0.01 (−0.43,0.42)	0.98	−0.26 (−0.8,0.28)	0.35	−0.15 (−0.69,0.4)	0.60	−0.14 (−0.69,0.42)	0.63
Multiple race	−0.07 (−0.46,0.31)	0.70	0 (−0.32,0.32)	0.99	−0.01 (−0.33,0.31)	0.96	−0.23 (−0.65,0.19)	0.29	−0.12 (−0.48,0.23)	0.50	−0.13 (−0.49,0.23)	0.48
Native Hawaiian or other Pacific Islander	−0.51 (−0.25,0.22)	0.17	−0.46 (−0.14,0.22)	0.18	−0.44 (−0.12,0.23)	0.20	−0.85 (−1.78,0.09)	0.07	−0.76 (−1.71,0.19)	0.11	−0.73 (−1.68,0.22)	0.13
Other Race	−0.07 (−0.47,0.34)	0.75	−0.06 (−0.4,0.27)	0.70	−0.05 (−0.38,0.28)	0.77	−0.17 (−0.56,0.23)	0.40	−0.09 (−0.44,0.26)	0.60	−0.08 (−0.43,0.27)	0.67
Maternal ethnicity	0.02 (−0.25,0.28)	0.89	0.1 (−0.13,0.33)	0.38	0.1 (−0.12,0.33)	0.37						
Marital status	0.18 (−0.15,0.51)	0.29	0.06 (−0.23,0.34)	0.69	0.05 (−0.23,0.34)	0.71						
Less than high school	−0.56 (−1.14,0.02)	0.06	−0.34 (−0.78,0.09)	0.12	−0.36 (−0.79,0.08)	0.11	−0.03 (−0.57,0.51)	0.91	0.07 (−0.38,0.51)	0.77	0.05 (−0.4,0.49)	0.83
High school degree, GED or equivalent	−0.53 (−0.96,−0.1)	0.02	−0.43 (−0.8,−0.06)	0.02	−0.45 (−0.82,−0.08)	0.02	0.06 (−0.38,0.49)	0.80	0.14 (−0.29,0.57)	0.53	0.11 (−0.32,0.53)	0.62
Some college/no degree/AA	−0.21 (−0.51,0.08)	0.16	−0.15 (−0.41,0.1)	0.23	−0.16 (−0.41,0.1)	0.22	0.21 (−0.1,0.52)	0.18	0.18 (−0.1,0.46)	0.20	0.18 (−0.1,0.46)	0.22
Mater’s, Professional or Doctorate Degree	0.03 (−0.15,0.2)	0.75	0.05 (−0.11,0.21)	0.52	0.05 (−0.11,0.21)	0.56	0.11 (−0.11,0.33)	0.32	0.09 (−0.11,0.29)	0.38	0.08 (−0.12,0.28)	0.42
<$30,000	−0.18 (−0.62,0.26)	0.42	−0.2 (−0.56,0.17)	0.29	−0.2 (−0.56,0.16)	0.27	−0.15 (−0.56,0.26)	0.46	−0.17 (−0.55,0.2)	0.36	−0.18 (−0.55,0.19)	0.34
$30,000–49,999	−0.1 (−0.43,0.24)	0.57	0.02 (−0.3,0.34)	0.88	0.01 (−0.31,0.33)	0.94	−0.21 (−0.57,0.15)	0.24	−0.05 (−0.44,0.35)	0.81	−0.06 (−0.45,0.33)	0.75
$50,000–74,999	−0.25 (−0.59,0.09)	0.15	−0.25 (−0.57,0.07)	0.12	−0.27 (−0.6,0.05)	0.10	−0.12 (−0.5,0.26)	0.53	−0.1 (−0.49,0.28)	0.60	−0.12 (−0.5,0.27)	0.55
$75,000–99,999	−0.01 (−0.28,0.26)	0.97	−0.04 (−0.31,0.22)	0.74	−0.05 (−0.31,0.21)	0.69	−0.06 (−0.42,0.3)	0.75	0.03 (−0.31,0.37)	0.85	0.02 (−0.32,0.36)	0.90
**Child Characteristics**												
Children sex	−0.34 (−0.5,−0.18)	0.00	−0.36 (−0.5,−0.21)	0.00	−0.36 (−0.5,−0.21)	0.00						
Birth weight	0 (0,0)	0.69	0 (0,0)	0.79	0 (0,0)	0.83	0 (0,0)	0.30	0 (0,0)	0.66	0 (0,0)	0.62
Children pre-pandemic life stage	−0.23 (−0.07,−0.39)	0.00	−0.23 (−0.09,−0.38)	0.00	−0.23 (−0.09,−0.38)	0.00	−0.23 (−0.06,−0.41)	0.01	−0.14 (−0.03,−0.31)	0.11	−0.14 (−0.03,−0.31)	0.10
Preterm status							−0.48 (−0.81,−0.15)	0.00	−0.25 (−0.55,0.05)	0.10	−0.24 (−0.54,0.06)	0.11
Gestational age							−0.01 (−0.04,0.02)	0.52	−0.02 (−0.06,0.02)	0.27	−0.02 (−0.06,0.02)	0.28
Children’s ethnicity							−0.16 (−0.41,0.09)	0.20	0.03 (−0.2,0.27)	0.79	0.03 (−0.21,0.26)	0.82

Notes. OR = Odds Ratio and CI = Confidence Interval; OR and CI values indicated in bold presents those with significance values greater than 0.008. References groups for categorical variables were White (maternal race), college degree (maternal education), Hispanic (maternal ethnicity), female (child’s sex), and infancy (pre-pandemic stage). All models were adjusted for pre-pandemic ASQ z-scores and significant covariates (*p*-values < 0.2).

## Data Availability

The data and procedures to replicate theses analyses are available upon request.

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
