# Peer review of "Effects of COVID-19 Financial and Social Hardships on Infants’ and Toddlers’ Development in the ECHO Program"

_ijerph, 2023, doi:10.3390/ijerph20021013_

Round 1

Reviewer 1 Report

Overall, this is an interesting paper on an important topic using a powerful set of data. 

Abstract :

- First sentence talks about children born during the Covid-19 pandemic but the study population is not about children born during the Covid-19 pandemic. I'm also not quite convinced of the sentence - why are they at risk?

- Development is vaguely defined. Please specify if you mean language, socioemotional domains, etc. Or all of them!

- Conclusion does not quite match the conclusion in the paper. The negative finding that pandemic-related hardships not associated with children's development is quite powerful actually.

Introduction:

The introduction would be strengthened with a focus on early years right from the start (it currently starts in paragraph 2). The age group of this study does not seem to be directly affected by closure of schools/remote learning. There is some limited data showing that parents being at home more was actually positive for early childhood (work from home). It would be helpful to balance the effects of the pandemic to not just be negative. 

Focusing on early childhood would also build a stronger conceptual model explaining why these pandemic hardships might have specific effects on early childhood. Many of the pandemic effects may be mediated by parental stress. I like how the introduction specifically discusses cognitive and socioemotional - why cognitive? For example, they are not yet in school. Is it because of parental stress and they are less able to spend time reading? Or are they able to spend more time reading?

Methods:

Study population would be strengthened with a little rationale about why 18 months before and after March 2020. Does this have to do with Covid-related changes 18 months afterwrads or a time period for developmental changes to show?  For readers 10 years from now, a detail about why march 2020 was chosen may be helpful to their memories. 

Measure

Primary Exposures: This section would benefit from more detail about why these measures were chosen (again, for our future readers 10 years from now who won't remember as well). The CEFIS is a well known covid survey, perhaps could identify which measures are also in the CEFIS? Or were these hardships ones that were highlighted in the news or in other literature? (unemployment rates, job loss). 

Discussion:

Paragraph 2 - I am not sure how male vulnerability in particular leads to specifically telehealth developmental supports. It may be that in person assessments may be even more important in early ages - and that pediatric primary care, where children early in age still come in person for visits may be a place to bundle these assessments. 

I find the negative finding quite powerful, though I would be sure to add "not associated with children's development as measured by the ASQ" to the conclusion.

Another interpretation may be that the measures of "pandemic-related disruption" should be reconsidered and perhaps another kind of disruption would be associated with developmental changes in early childhood. Again would also need to support why the measures are truly pandemic related. There's also the question of whether getting covid may affect children (long covid studies that are still ongoing). given that the paper is discussing pandemic effects, should mention potential effects of the disease itself.

National test results recently showed that there were significant math losses in children during the pandemic (https://www.nytimes.com/2022/09/01/us/national-test-scores-math-reading-pandemic.html?). Perhaps we could strengthen the context of these findings by talking about how early childhood developmental needs are different. 

Finally, it the emphasis on service delivery feels misplaced. Most children with mild developmental delay do not qualify for services and the effect sizes here are mostly small. The first steps in a pediatrician office are really to engage the parents in stimulating their child's development and monitoring closely. It feels more as if screening would be more important. 

Author Response

Reviewer 1:

Overall, this is an interesting paper on an important topic using a powerful set of data. 

Abstract :

- First sentence talks about children born during the Covid-19 pandemic but the study population is not about children born during the Covid-19 pandemic. I'm also not quite convinced of the sentence - why are they at risk?

- Thank you for this comment. We have changed the first sentence to provide more backgrounds on the purpose of study and contributions to the current literature on this topic. Specifically, the first sentence has been changed to reflect that we need more research to examine development and trajectories in association with experienced of COVID-19 hardships during early years.  

“The financial hardships and social isolation experienced during the COVID-19 pandemic have been found to adversely affect children’s developmental outcomes. While many studies thus far have focused on school-aged children and the pandemic-related impacts on academic skills and behavior problems for older children, relatively less is known about pandemic hardships and associations with children’s development during early years.”

- Development is vaguely defined. Please specify if you mean language, socioemotional domains, etc. Or all of them!

- We have revised the sentence to include that while we examined all aspects of development, our particular focus was on communication and socioemotional development

“Using a racially and economically diverse sample, we examined whether hardships experienced during the pandemic were associated with children’s development with a particular focus on communication/language and socioemotional development.”

- Conclusion does not quite match the conclusion in the paper. The negative finding that pandemic-related hardships not associated with children's development is quite powerful actually.

- We have revised the first sentence of our conclusion to include lack of significant findings for the associations between pandemic-related hardships and children’s development.   

Negative developmental changes from pre- to during-pandemic were found in boys, yet we did not find any associations between increased experience of pandemic-related hardships and children’s development.”

Introduction:

The introduction would be strengthened with a focus on early years right from the start (it currently starts in paragraph 2). The age group of this study does not seem to be directly affected by closure of schools/remote learning. There is some limited data showing that parents being at home more was actually positive for early childhood (work from home). It would be helpful to balance the effects of the pandemic to not just be negative. 

_ we would like to thank you for this comment and the importance of discussing potential positive impacts of pandemic and preventive guidelines, particularly for young children. We have added the following sentences to the first paragraph:

“Positive impacts also have been noted as the result of preventive guidelines such as increase in time that family spend with each other and hence bonding among family members as well as increase in time that children spend in nature which has been associated with positive impacts on development. The increase in bonding time that parents can have with their children as the result of working from home practices can be particularly important for young children as the first three years are important in development of attachment

Focusing on early childhood would also build a stronger conceptual model explaining why these pandemic hardships might have specific effects on early childhood. Many of the pandemic effects may be mediated by parental stress. I like how the introduction specifically discusses cognitive and socioemotional - why cognitive? For example, they are not yet in school. Is it because of parental stress and they are less able to spend time reading? Or are they able to spend more time reading?

  • We now have added a couple of sentences in the introduction to describe why we were particularly interested in communication and socioemotional development

“Given the high dependency that children have on caregivers during the first few years of life, they may be more vulnerable to parental stress and negative family dynamics, which may impact the quality and frequency of positive parent-child interactions.” Thus, we were particularly interested in evaluating the associations between pandemic-related hardships and young children’s communication and socioemotional development.

Methods:

Study population would be strengthened with a little rationale about why 18 months before and after March 2020. Does this have to do with Covid-related changes 18 months afterwrads or a time period for developmental changes to show?  For readers 10 years from now, a detail about why march 2020 was chosen may be helpful to their memories

  • Thank you for this comment. In the revised version of manuscript, we have revised the sentence on page 3 to clarify our rationale for choosing the 18-month time window.

“We only considered the pre-pandemic ASQ data that were collected within the 18-month window prior to March 2020 (i.e., August 2018-March 11th 2020) to match the time frame for the ASQ data included in the analysis from the during-pandemic period (March 2020-August 2021).”  

Measure

Primary Exposures: This section would benefit from more detail about why these measures were chosen (again, for our future readers 10 years from now who won't remember as well). The CEFIS is a well known covid survey, perhaps could identify which measures are also in the CEFIS? Or were these hardships ones that were highlighted in the news or in other literature? (unemployment rates, job loss)

Thank you for this comment. More detailed information about the development process of COVID-19 survey is provided on page 3 of revised manuscript. The COVID-19 measure used in the current study was developed by the by the PIs of cohort studies participating in the ECHO Consortium in April 2020.  The items used to capture financial and social hardships experienced during the COVID-19 pandemic were similar to those in CEFIS (e.g., items 16-18 in the CEFIS-parent report measured financial hardships and reflected the temporary or permanent job loss or reduced work hours for either parent).

“The ECHO COVID-19 survey was developed in April 2020 by a group of ECHO investigators with various expertise (e.g., clinical and developmental psychologists, epidemiologists) using existing surveys and source materials from leading organizations. More detailed information about the development process can be found in Blackwell et al. (2022) article. “

“The items used to capture financial and social hardships were similar to those from other well-established COVID-19 survey such as the COVID-19 Exposure and Family Impact Survey (CEFIS)”

Discussion:

Paragraph 2 - I am not sure how male vulnerability in particular leads to specifically telehealth developmental supports. It may be that in person assessments may be even more important in early ages - and that pediatric primary care, where children early in age still come in person for visits may be a place to bundle these assessments

  • We agree with your comment that during these early years when children are required to see their pediatricians more frequently, it may be easier to integrate the early developmental screening into the care routine more effectively and systematically.
  •  

“These results also highlight the importance of improving service deliveries and early developmental screening during the well-child care visits at the pediatric care offices particularly at the time of adversities to ensure that young children who are at-risk including male children are connected to intervention and treatment services.”

I find the negative finding quite powerful, though I would be sure to add "not associated with children's development as measured by the ASQ" to the conclusion.

  • Thank you for this suggestion. The sentence has been revised.

Another interpretation may be that the measures of "pandemic-related disruption" should be reconsidered and perhaps another kind of disruption would be associated with developmental changes in early childhood. Again would also need to support why the measures are truly pandemic related. There's also the question of whether getting covid may affect children (long covid studies that are still ongoing). given that the paper is discussing pandemic effects, should mention potential effects of the disease itself.

National test results recently showed that there were significant math losses in children during the pandemic (https://www.nytimes.com/2022/09/01/us/national-test-scores-math-reading-pandemic.html?). Perhaps we could strengthen the context of these findings by talking about how early childhood developmental needs are different

  • We appreciate these thoughtful comments and suggestions. In the revised manuscript, we have added a paragraph to link the results of this study to results from studies conducted among school-aged children and have provided some directions for future research based on this comparison. Specifically, we have discussed how differences in developmental needs across various developmental stages may explain differences in results and why future studies may need to focus on other aspects of pandemic effects such as impacts on family dynamics, important for early development, as well as the disease itself using a longer follow-up.

“Many studies in the U.S. have reported learning losses in the areas of reading and math from pre- to post- pandemic among school-aged children [26-28]. The results of a national study also showed that the achievement gaps in the areas of reading and math between students in the low and high poverty schools significantly increased from pre – to post- pandemic years (difference of .10-.20 SDs) [26]. In the current study, we did not find any associations between financial and social disruptions caused by the pandemic and infants’ and toddlers’ development, measured by an early screening tool. Given that early childhood developmental needs are different from those from older and school-aged children, future research may consider focusing on other aspects of pandemic-related disturbances such as the pandemic effects on family dynamics, parent-child interactions and parental stress. Additionally, more research is warranted to examine the effects of pandemic-related hardships and COVID-19 infection on children’s later developmental outcomes using a longer follow-up. The difference in the impact of disruptions related to the pandemic for younger and older children may also reflect the possibility that classroom-based learning platforms and achievement indices in the U.S. are more sensitive to disruptions than learning that takes place in the home environment. If further research studies find these differences, there may be a need to revise our traditional learning modes in the school context, which may lead to greater flexibility and adaptability to future hardships and disruption both on a national scale, as in the case of COVID, and on a state, community, and even individual student level. “ 

  • Finally, it the emphasis on service delivery feels misplaced. Most children with mild developmental delay do not qualify for services and the effect sizes here are mostly small. The first steps in a pediatrician office are really to engage the parents in stimulating their child's development and monitoring closely. It feels more as if screening would be more important
  • The last sentence in the conclusion has been revised to reflect your suggestion. Thank you.

Reviewer 2 Report

The manuscript presents results from a longitudinal cohort of children in the US. The outcome measures (ASQ scores) are high quality, validated tools for this construct. As noted in the paper, the strength of longitudinal data is very useful for the field. Given the international audience, I suggest adding "in the United States" to the 2nd sentence in the introduction. In table 1, I was confused as to why the % symbol was only included in the first column, when it applied across the table. 

Author Response

The manuscript presents results from a longitudinal cohort of children in the US. The outcome measures (ASQ scores) are high quality, validated tools for this construct. As noted in the paper, the strength of longitudinal data is very useful for the field. Given the international audience, I suggest adding "in the United States" to the 2nd sentence in the introduction. In table 1, I was confused as to why the % symbol was only included in the first column, when it applied across the table. 

  • We appreciate your positive comments on our manuscript. Changes requested are implemented in the second sentence of introduction and in Table 1. Thanks for bringing them to our attention.

Round 2

Reviewer 1 Report

Nice job improving the framing of the study question and including more detail in the methods. Appreciate the value of null results and nicely presented within the context of prior work.